# The role of references and the elusive nature of the chemical bond

Ángel Martín Pendás [1] & Evelio Francisco[1]

Chemical bonding theory is of utmost importance to chemistry, and a standard paradigm in which quantum mechanical interference drives the kinetic energy lowering of two approaching fragments has emerged. Here we report that both internal and external reference biases remain in this model, leaving plenty of unexplored territory. We show how the former biases affect the notion of wavefunction interference, which is purportedly recognized as the most basic bonding mechanism. The latter influence how bonding models are chosen. We demonstrate that the use of real space analyses are as reference-less as possible, advocating for their use. Delocalisation emerges as the reference-less equivalent to interference and the ultimate root of bonding. Atoms (or fragments) in molecules should be understood as a statistical mixture of components differing in electron number, spin, etc.

[1] Departamento de Química Física y Analítica, Universidad de Oviedo, 33006 Oviedo, Spain. ✉email: ampendas@uniovi.es

Coinciding with the centennial commemoration of the electron-pair bond proposed by Gilbert Newton Lewis in 1916[1], a considerable number of research papers[2,3], perspective articles[4–6] and books[7,8], all revisiting the nature of the chemical bond, have appeared in the literature, showing the vitality of the field. Most of these recent accounts[4] agree upon the framework forged by K. Ruedenberg and coworkers in the early 1960s. Following Hellmann's suggestion[9,10], this standard model identifies the kinetic energy lowering suffered by the interacting moieties, triggered by constructive interference of their wavefunctions, with the driving force behind bond formation. This view opposes that of Slater[11], who used the virial theorem to vindicate the role of the potential energy and the accumulation of electron density in the internuclear region upon bonding, and also in part that of Feynman[12], who developed an image of bonding in terms of electrostatic forces. As Frenking and coworkers have emphasized[4], it is essential that on top of this well accepted physical image we build chemical bonding models that translate the physical language into the fuzzy, but predictive, chemical concepts. Although different schools still diverge on how fine details are interpreted and, for instance, valence bond (VB) advocates frequently collide with pure molecular orbital (MO) practitioners (see, e.g., refs. [13–16] for recent controversies), a minimal set of points, that include Hellmann's and Ruedenberg quantum mechanical interference, seems to have been agreed upon. According to this view, it is the interference among the different wavefunctions of the fragments which are forming a chemical bond that leads to a general decrease in the kinetic energy of the system, driving it towards equilibrium.

Here we show that even these accepted points necessarily imply the choice of both internal (state) or external (energetic) references which bias interpretations. We examine the nature and consequences of those biases and show how the consideration of interacting atoms or molecules as objects in real space minimizes the reference bias as much as possible. By introducing this real space picture we show that it is electron delocalization that underpins chemical bonding.

## Results and discussions

Ruedenberg and coworkers[7,17,18] have pointed out that building a theory of chemical bonding requires as a first, absolutely necessary prerequisite to postulate that atoms, or larger entities if necessary, are somehow preserved in molecules. Chemical bonds occur among interacting moieties that we must single out from the final stable or metastable molecular arrangements. Since quantum mechanics is an intrinsically non-separable theory, how these atoms are introduced and manipulated provides a very first source of bias.

**The wavefunction reference bias in the $H_2^+$ molecular ion.** Take the simplest one electron $H_2^+$ ion, described under the clamped-nuclei approximation by a wavefunction $\Psi$[19]. Figure 1 shows a potential energy curve with several energetic decompositions that will be used in the following. Ruedenberg's variational reasoning is relentless. To lower its energy, the electron thrives to balance the opposing kinetic and potential energy demands. The first is lowered with electron delocalization or dilution, while the latter becomes more negative with localization around the nuclei. The electron is thus pulled towards the nuclei as much as the kinetic resistance (also called kinetic pressure) allows. Kinetic ($T$) and potential energy ($V$) lowering dominate the long-range and short-range behavior, respectively, and at equilibrium, the variational balance leads to fullfilment of the virial theorem, and $E = -T$.

Further chemical analysis requires establishing a reference, which is standardly taken as the H atom and its exact one-electron states (Supplementary Note 1), and the molecular wavefunction is recast in terms of a set of so-called quasi-atomic orbitals (QUAOs, $\phi_{a,b}$), which are then compared to the isolated $\phi_{1s}$ function and used to provide an exact energy and density decomposition. Squaring the $\Psi$ amplitude leads to a sum of the squares of the quasi-atomic densities and to an interference term: $\rho = \phi_a^2 + \phi_b^2 + \rho_I$. It is found that the QUAOs pass from slight expansion at large $R$ to significant contraction at equilibrium, and that it is only the interference $T_I$ lowering which drives the system to an equilibrium geometry, the accumulation of density in the internuclear region being also entirely caused by interference. In this standard model, it is the constructive interference of the atomic functions that allow the electron to dilute or delocalize. Many researchers have contributed to the details of this image over the years[20–25].

What is not usually contemplated in general treatments is: (i) that the correct $R \to \infty$ asymptotic reference of $H_2^+$ is a spatially entangled state and not the broken symmetry H+ H$^+$ one, (ii) that the analysis of the effect of interference depends on spatially separated basis functions, introducing completely spurious two-center terms in an otherwise one-particle problem. As the first point is regarded, it is all but clear that the infinite distance asymptotic wavefunction $\Psi = 1/\sqrt{2}(\phi_{1sa} + \phi_{1sb})$ describes an entangled electron with a 50% probability of being found around each nucleus, $p(n_a = 1, n_b = 0) = p(n_a = 0, n_b = 1) = 1/2$, and not a H atom and a proton, $p(n_a = 1, n_b = 0) = 1, p(n_a = 0, n_b = 1) = 0$. This takes us to R. Feynman[26], who interpreted the $H_2^+$ system in terms of the dynamic flip–flop jump of the electron between the two atomic regions (Fig. 1, right panel). In the long distance limit a state that is initially described by the localized $\phi_a$ function will tunnel to $\phi_b$ with a rate $\tau = (\pi \hbar)/\Delta E$, where $\Delta E$ is the system's first excitation energy. As distance increases and $\Delta E$ tends to zero, the tunneling time grows indefinitely. This dynamic picture has been exploited over the years by Nordhom and Backsay[27,28], among others.

As it stands out, interference is an a posteriori result of the wavefunction decomposition, not an intrinsic feature of the system. $H_2^+$ is a problem of a particle moving in a potential $V(\mathbf{r})$. Isolating two wells (the two nuclei) is necessary for chemical analyses, but alien to the physics of the problem. Let us consider a textbook one-dimensional particle moving in a box of length $L$ resulting from the interpenetration of two smaller left and right boxes of length 2/3$L$, which we label $a$ and $b$ (see Fig. 2). After eliminating the right-$a$ and left-$b$ constraining walls (dashed lines at $2L/3$ and $L/3$, respectively), we build an approximation to the solutions of the larger box in terms of those of the original $a$ and $b$ problems. With the sinusoidal ground states $\phi_a$, $\phi_b$ of the initial boxes we can approximate the ground state of the final system as $\Psi = N(\phi_a + \phi_b)$. The ground state of the large box can be recast as the constructive combination of two symmetric, deformed solutions of the $a, b$ boxes. For an observer unaware of our gedanken small boxes, using the solutions of these two hidden ancillary systems is a completely arbitrary process, as it is assigning any physical sense to the constructive interference of $\phi_a$ and $\phi_b$. This internal bias soaks the standard model of chemical bonding. As $V = 0$ inside the box, $E = T$, and it suffices to consider $\hat{T} = -(1/2)d^2/dx^2$. All the following considerations apply also to $V$ when it is non-vanishing. Recall that (in atomic units) the energy of the ground state of an electron in a box of size $L$ is $\pi^2/(2L^2)$. By releasing the right-wall constraint at $2L/3$ of an electron initially confined in the left $a$ box and letting it occupy the $a \cup b$ larger box, $E = T$ decreases. The role of the release of spatial constraints in chemical bonding has been put forward several times[28], and in our opinion is the true core upon which all chemical bonding treatments rest. In terms of interference,

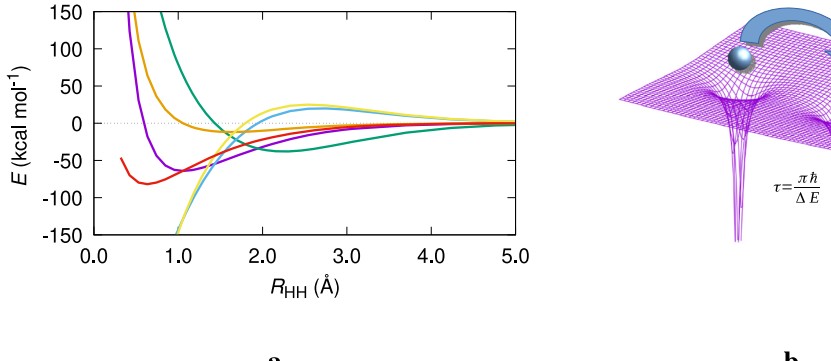

**a** **b**

**Fig. 1 Bonding in $H_2^+$. a** FCI cc-pVTZ potential energy curve of $H_2^+$ along with several of the energetic descriptors obtained by decomposing the space in two symmetric regions associated to the $a$ and $b$ nuclei. All energies referred to their infinite distance limit, in a.u. $E$ (purple), $T$ (green), and $V$ (cyan), stand for the full system's total, kinetic, and potential, energies, respectively. $E_{int}$ (orange) is the sum of the internuclear repulsion energy and the attraction energy of the electron density contained in the $a$ region with the $b$ nucleus, and vice versa. Similarly, $V_{en'}$ (yellow) is the sum of the attraction energy between the electron density in $a$ with nucleus $a$ and its $b$ equivalent. Finally, $E_{self}$ (red) is the sum of the self-energies of the two regions, see the text, where it is also explained that in the broken symmetry spatial interpretation, $V_{en'}$ and $E_{self}$ can also be interpreted as those of a non-entangled H atom interacting with a proton. **b** Illustration of Feynman's dynamic picture of the 2c,1e bond. $\tau$ and $\Delta E$ are the inter-well tunneling rate and the system's first excitation energy, respectively.

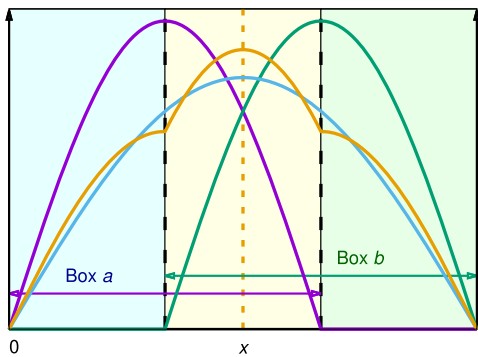

**Fig. 2 A model for the wavefunction reference bias.** A one-electron one-dimensional box of length $L$ built from the interpenetration of two smaller left ($a$, blue) and right ($b$, green) boxes of length $2/3L$. The interpenetration region is depicted in yellow. The ground states of an electron initially confined in either the $a$ or $b$ boxes, $\phi_a$, $\phi_b$ are shown as the purple and green solid curves. If the impermeable walls (dashed black lines) are released, the electron delocalizes, and its ground state, $\Psi$ is depicted in cyan. When $\Psi$ is approximated by the gerade linear combination of $\phi_a$ and $\phi_b$, $\Psi = N(\phi_a + \phi_b)$, in orange, $S_{ab} \approx 0.2153$. The spatial partitioning of the large box into two equivalent non-interpenetrating regions of length $L/2$ is depicted by the short-dashed orange line.

$T = T_a + T_b + T_I$ where $T_I = -(1/2)\int_0^L \phi_a(x)(d^2\phi_b(x)/dx^2)dx$. Notice that in $H_2^+$ $T_I$ is interpreted as an interatomic term. This is an artifact coming from the mixing of $\phi_a$ and $\phi_b$. $T$ is a one-particle, thus one-center observable. For the final ground state, $T = (-1/2)\int_0^L \Psi(x)(d^2\Psi(x)/dx^2)dx$, and no two-center contributions arise. It is only when we decompose $\Psi(x)$ at point $x$ into two contributions assigned to two different centers that an inter-center interference term appears. We conclude that two-center contributions for one-particle operators should be avoided to bypass internal or state reference biases. They should only be present when considering the interelectron repulsion, a true two-particle operator that can display two-center behavior. Figure 2 helps us envisage how we can still associate the particle to two centers (boxes) without invoking any internal reference or interference term. Since the initial limits of the boxes disappear when the wall constraints are released, a reasonable assumption follows: if the electron is in the left/right region of the final box we

associate it to the remnants of the left/right initial boxes. Symmetry arguments put the separation of the in-the-molecule left and right regions at the short-dash orange line in the figure. A natural spatial partitioning that divides space into (chemically) relevant regions appears.

No internal reference or arbitrary choice remains at all. In more general cases, a choice about how the spatial partitioning is performed is needed, but the number of degrees of freedom for this decision is much smaller than in other cases. The spatial partition leads to a probability of finding the electron in each of the boxes-in-the-box equal to $1/2$, and to a regional kinetic energy $T^{a,b} = T/2$. We can still use the original $a$, $b$ boxes as references, and compute the changes in $T^a$ on going from the isolated $a$ box to the $a$ box-in-the-box situation. This is the basis of real space bonding analyses. In this view, an electron found in a given (chemical) region is never considered part of another one, as when interpenetration is allowed.

A spatial partitioning of the entangled ground state of $H_2^+$ leads to a 50% probability of the electron being found in any of the two atomic regions (i.e., basins), at any distance: $p(n_a = 1, n_b = 0) = p(n_a = 0, n_b = 1) = 1/2 = \int_a \rho(\mathbf{r})d\mathbf{r}$. All kinetic and electron-nuclear potential energy terms can be decomposed into basin contributions: $T = T^a + T^b$, $V_{en} = V_{en}^{aa} + V_{en}^{ba} + V_{en}^{ab} + V_{en}^{bb}$, with $V_{en}^{ab}$ being, for instance, the electron–nucleus (en) attraction between the electron, when it is contained in region $a$, and nucleus $b$. There are powerful reasons to choose the atomic basins provided by the quantum theory of atoms in molecules (QTAIM)[29], a standard which we will follow. The sum of $T_a + V_{ne}^{aa}$ corresponds to the energy of the electron residing in atom $a$, that is usually called its self-energy, $E_{self}^a$. In polyelectronic cases it also includes the interelectron repulsion of the electrons located in $a$, $V_{ee}^{aa}$. Similarly, the sum $E_{int}^{ab} = V_{en}^{ab} + V_{en}^{ba} + 1/R_{ab}$ (with an additional polyelectronic $V_{ee}^{ab}$ repulsion if needed) is the interaction of the electron when residing in a basin with the other nucleus and vice versa, plus the internuclear repulsion $V_{nn}^{ab} = 1/R_{ab}$. It is called the interatomic interaction energy. Adding up, $E = E_{self}^a + E_{self}^b + E_{int}^{ab}$. This is the so-called interacting quantum atoms (IQA) energy decomposition[30–32]. All of these components can be found in Fig. 1 (see also Supplementary Notes 2, 3), where $V_{en}^{aa} + V_{en}^{bb}$ is called $V_{en'}$. $E_{self}^a$ tends to $-1/4$ a.u. at dissociation, which is obviously equal to $p(1,0) \times E(H) + p(0,1) \times E(H^+)$. $E_{self}^a$ is

stabilized as $R$ decreases, contrarily to its equivalent $E_{intra}$ in Ruedenberg's analysis, since the strongly stabilizing interference kinetic energy $T_I$ of the latter is now an intra-basin phenomenon. Interestingly, even though no atomic functions are used in this real space analysis, orbital-like expansion and contraction can be easily recognized from how $V_{en}^{aa}$ (or $V_{en'}$ in the figure) changes with respect to their values at dissociation. This change is initially positive to become negative as $R$ decreases. The contrary is true for $\Delta T^a$. Notice also how $V$ differs only slightly from $V_{en'}$ for a wide range of distances. The real space viewpoint unveils the release of constraints, thus delocalization, as the ultimate root of chemical binding. If we confine the electron to either the left or right basins, variational reasoning leads to $E_{self}^a > E(H)$ at any distance. Once electron delocalization (basin hopping á la Feynman) is allowed, chemical bonding sets in. The real space analysis also permits a broken symmetry perspective (Supplementary Notes 2 and 3).

The evolution of one-electron bonds towards polarity can be followed in the alkali dimer cations, $AB^+$, which are well modeled by a 2c, 1e paradigm. A single reference (SR) wavefunction for $LiNa^+$ leads to $p(n_{Li} = 3, n_{Na} = 10) = 0.748$, $p(2, 11) = 0.251$, with a residual contribution for other electron counts: a polar 2c,1e bond where the electron has 25% probability of being found in the Na atom. Manipulating the difference of electronegativities between the atoms induces a shift from the (close to) perfect non-polar 2c,1e bond in $Li_2^+$ with $p = 1/2$, passing through $p \approx 0.75$ in $LiNa^+$, towards $p \approx 1$ in $LiCs^+$.

**The $H_2$ molecule**. Adding a second electron to the $H_2^+$ system leads to the electron-pair bond. As in $H_2^+$, it is the kinetic energy interference term, $T_I$, which drives bonding in the standard model. Nevertheless, a couple of new insights, which are usually hidden or not fully considered, stand out: (i) An important contribution to the interatomic potential, the so-called coulombic sharing[33] term appears because of the existence of intra-atomic electron–electron repulsion. (ii) The electron-pair bond is not a result of electron spin pairing. Actually, the binding energy of dihydrogen is smaller than twice that of $H_2^+$ (109.5 versus 64.4 kcal mol$^{-1}$),[34] and $H_2$ should be viewed as two independent two-center one-electron bonds, slightly weakened by the inter-electron repulsion that is absent in $H_2^+$[4]. According to Ruedenberg and coworkers, this picture is general, and kinetic interference is the root of all bonding. Head-Gordon et al.[35–37] have questioned this view, pointing to electron delocalization as the most general bonding driver. Care has to be taken, anyway, since some of these arguments were based on using the triplet state of the $H_2$ molecule as a reference.

In the real space picture of $H_2$[30], the spatial regions associated to the H atoms are symmetry determined. The self-energy of each atom $E_{self}^{a,b}$, increases upon molecular formation, so that all binding comes from $E_{int}^{ab}$, which is naturally decomposed into a repulsive ionic $V_{cl}^{ab}$ term (net charge dominated) and an attractive covalent-like $V_{xc}^{ab}$ (exchange-correlation) one (see Supplementary Note 2). A vital point stands out: the two electrons can be both found in one of the two atoms $a$ or $b$. Atoms in molecules should not be imagined possessing a fixed number of electrons. If this is done, we face an electron number bias, for we must choose arbitrarily the number of electrons of our references. In $H_2$, three possible electron distributions are possible: (1, 1), (2, 0) and (0, 2), labeled according to the number of electrons in $a$ and $b$. The first is compatible with neutral atomic references, while the other two correspond to ionic $H^-$ and $H^+$ ones. An interacting atom or fragment $a$ is in a mixed open quantum state, with a probability $p(n_a)$ of being found with a given electron count $n_a$. In this view,[38,39] $E = \sum_i p_i E_i$. In the simplest MO $1\sigma_g^2$ wavefunction of $H_2$,

$p(1, 1) = 1/2$, and $p(2, 0) = p(0, 2) = 1/4$ (see Supplementary Note 2)[39]. This describes two independent 2c,1e $H_2^+$-like bonds, each electron ignoring the other, with a 50% probability of being found in each atom.

The role of correlation is also clear. A multireference (MR) wavefunction in $H_2$[40] shows that $p(1, 1) = 0.584$, $p(2, 0) = p(0, 2) = 0.208$ at $R = 1.4$ a.u., corroborating that electron correlation decreases delocalization and increases the probability of finding each of the two electrons closer to the (different) nuclei. As $R$ increases, $p(1, 1)$ tends to 1, and at $R = 6.0$ a.u., it is equal to 0.988 (full configuration interaction results with larger basis sets[38,41] do not alter this image). The energy of the (1, 1) electron distribution at $R = 1.4$ a.u. is −1.260 a.u., with $E_{self}^a = -0.653$ and $E_{int}^{ab} = +0.047$ a.u., respectively. These tend to −0.500 and 0.000 a.u. at dissociation, respectively. As the (2, 0) distribution is regarded, at $R = 1.4$ a.u. $E = -0.996$ a.u., with $E_{self}^a = -0.496$, $E_{self}^b = 0$, and $E_{int}^{ab} = -0.500$ a.u., respectively. These tend to −0.467 (the energy of a hydride anion at the single-determinant level with the chosen basis set) and 0.000 a.u., respectively, at dissociation. These numbers show how grossly incorrect choosing references with fixed number of electrons can be. In each of the (1, 1) or $(2, 0) \equiv (0, 2)$ distributions a fixed number of electrons reside in each atomic region. When strictly one electron is found in each atom, all the interaction is quasiclassical (although built with the quantum mechanical density), and an electrostatic theorem[42] guarantees that their interaction is repulsive. In agreement with the standard (i.e., Ruedenberg) model, the very negative $(1, 1)E_{self}$'s imply a highly contracted electron density. Contrarily, for the (2, 0) $H^-$–$H^+$ distribution $E_{int}$ is large and negative, behaving as $-1/R$ asymptotically, while $E_{self}^{H^-}$ reveals a slightly contracted hydride. If delocalization is prohibited[43], the (1, 1) distribution would lead to a repulsive potential energy curve, for a minimization of the (1, 1) energy would lead to $E_{self}^a > E^H$ (and $E_{int}^{ab} > 0$).

Summarizing: (i) Associating interference as the driving force behind chemical bonding is, to a large extent, the result of assuming a set of internal references. Since it needs not be imposed, it cannot be taken as the final root of the chemical bond, which we ascribe to delocalization in its most general sense; (ii) Even for the simplest dihydrogen case, the consideration of a neutral reference disregards the fact that at equilibrium about half the time one of the H atoms bears two electrons.

**The energy reference bias**. Out of the many energy decompositions in the literature, Ziegler and Rauk's EDA[44], many times in combination with the natural orbital for chemical valence (NOCV) analysis[45] has proven useful in building chemical bonding models. We use it here for illustrative purposes. Its basic tenet is that building a chemical bonding model needs (at least) two fragments that interact, from which we measure or compute changes. These fragments are isolated at the equilibrium geometry of the molecule, and their electronic structure is determined in electronic states that may not coincide with their ground state, but that have chemical sense (vide infra). Then, their interaction energy $\Delta E_{int}$ is determined through a sequence of steps: (i) the classical electrostatic interaction between their total (nuclear and electronic) interpenetrating charge densities, $\Delta E_{elstat}$ is determined. This is typically stabilizing, and EDA-NOCV advocates have emphasized the role of these forgotten quasi-electrostatic terms in bonding[5]. (ii) The frozen fragments' wavefunctions are antisymmetrized, and the concomitant rise in energy is associated to the Pauli repulsion $\Delta E_{Pauli}$ among the non-relaxed electrons of the fragments; (iii) Finally, the antisymmetrized function is orbitally relaxed (i.e., a full calculation is done in the molecule). The energy drop is the orbital relaxation energy, $\Delta E_{orb}$, which is linked to covalency. In the end,

$\Delta E_{int} = \Delta E_{elstat} + \Delta E_{Pauli} + \Delta E_{orb}$. If needed, a dispersion contribution, $\Delta E_{disp}$ is also added. When the geometries, electronic or spin states of the fragments do not coincide with those at dissociation, another term, the preparation energy, $\Delta E_{prep}$, is necessary, so that $\Delta E = -D_e = \Delta E_{prep} + \Delta E_{int}$. The application of this EDA is not always viable: it requires final single determinant descriptions (or Kohn–Sham quasi-determinants) that hamper their application to systems where static correlation is important, and it sometimes requires tricks to build the appropriate molecular spin state from those of the fragments.

**Intrinsic bond strength and metastable bonds**. Given that $D_e$ includes $\Delta E_{prep}$ that can mask bonding effects, it has been suggested[2] that it is $\Delta E_{int}$ that provides a faithful descriptor of the intrinsic bond strength, the chemically relevant bond energy. Since this is obviously a troublesome indicator (e.g., in metastable molecules), many researchers, pioneered by Cremer and Kraka[46], prefer to abandon energies to measure bond strength and to turn to local measures, independent of references, such as local force constants. Although these are not uniquely defined, they have considerable advantages, and have been recently advocated by Zhao et al. as a solution to the bond dissociation energy (BDE) conundrum[47]. It has also been suggested that reference-less real space $E_{int}$'s can be used as in situ bond strength parameters[48]. We take as an example the $He_2^{2+}$ system. This is a metastable two-electron ion. A MR calculation yields a metastable minimum at $R = 1.332$ a.u., $201.2$ kcal mol$^{-1}$ above the dissociation limit. The well is about $34$ kcal mol$^{-1}$ deep, with a very large force constant of $12.51$ mdyn Å$^{-1}$, larger than in dihydrogen, $5.55$ mdyn Å$^{-1}$.[49] A real space partition at the metastable minimum yields a simple 2c,2e close to that in $H_2$: $p(1,1) = 0.617$, with a covalent bond order of $0.765$. $E_{int} = +266.4$ kcal mol$^{-1}$, partitioned into $462.2$ kcal mol$^{-1}$ electrostatic destabilization between the $He^+$ cations (the pure Coulombic repulsion between two unit point charges at that distance is $471.1$ kcal mol$^{-1}$) and $-195.8$ kcal mol$^{-1}$ exchange-correlation stabilization (Supplementary Note 4). The covalent component in dihydrogen is smaller, around $-149$ kcal mol$^{-1}$. Applying the EDA prescription to a single-determinant wavefunction obtained at the FCI geometry with $He^+$ references we obtain (all data in kcal mol$^{-1}$) $\Delta E_{elstat} = 474.3$, $\Delta E_{Pauli} = 0$, $\Delta E_{orb} = -229.5$. Since in a 2e singlet the EDA Pauli repulsion vanishes, $\Delta E_{orb}$ can be rather safely associated with the onset of covalency. The reference-less $E_{int}^{ab}$ together with its ionic and covalent components provide local energetic information about bond strength that can be used on a par with local force constants.

**Chemical bonding models from energy references**. Frenking and coworkers[5] have shown how to make the most of the a priori freedom to choose the electronic/spin states of the fragments. According to their protocol, different references correspond to different bonding models, and a variational-like argument is used to select the most appropriate fragment choice: the preferred model is that displaying the smallest $\Delta E_{orb}$ step, with fragments as prepared as possible for bonding to take place. There exist many successful examples of this approach. It allows to rationalize the planar and trans-bent geometries of $C_2H_4$ and $Si_2H_4$, respectively, and of many other $E_2X_4$ heavy molecules (see Fig. 3). The $CH_2$ fragments' ground state is a $^3B_1$ triplet, with two unpaired electrons perfectly suited to engage in a double bond. On the contrary, the heavier $EH_2$ fragments are $^1A_1$ ground state singlets, which have to be formally excited to the triplet to get involved in electron-sharing. Thus they prefer donor–acceptor interactions that lead to non-planar geometries[2]. $C_2F_4$ is a mostly interesting case: although the isolated $CF_2$ fragment is a singlet with a singlet–triplet gap of about $51.2$ kcal mol$^{-1}$, $\Delta E_{orb}$ is considerably smaller if the EDA

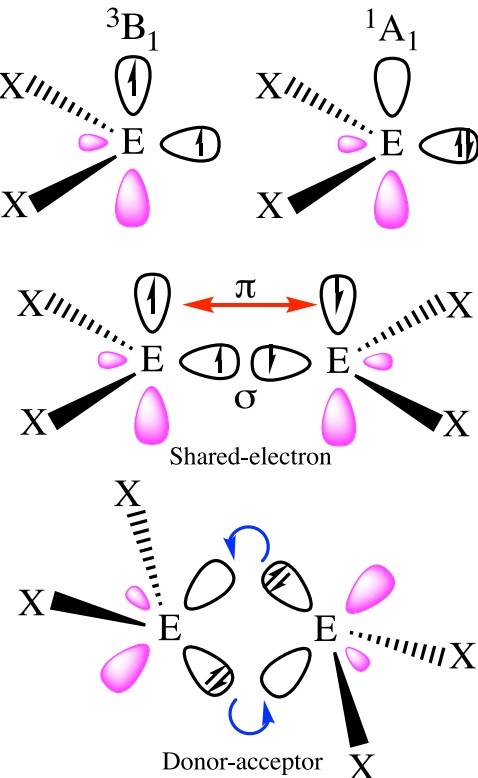

**Fig. 3 Transition from shared-electron to donor–acceptor chemical bonding models in $E_2X_4$ systems.** If fragments interact in their ground state throughout the full association path, planar or trans-bent geometries are expected for triplet or singlet $EX_2$'s, respectively. In the case of small singlet–triplet promotion energies, the bonding mode can change along the association, as in $C_2F_4$.

fragments are made to interact as triplets[50], so that the $C_2F_4$ ground state is, as in ethylene, planar and well described by a double bond. The singlet to triplet excitation cost manifests in the very small $D_e = 74.4$ kcal mol$^{-1}$, much smaller than $\Delta E_{int} \approx -200$ kcal mol$^{-1}$, of a normal C–C double bond. Association of two $^1A_1$ $CF_2$ moieties proceeds via a dative or donor–acceptor trans-bent geometry which is progressively planarized: bonding interactions may change during bond formation, warning us about using dissociation products as probes for dative or electron-sharing interactions.

*Single reference systems*. It is at this point that all our alarms ring together. In order to appropriately use the energetic promotion of fragments during bond formation one must consider their mixed quantum nature and allow for both electron number and electronic/spin state fluctuations. Moreover, electron reorganizations along a reaction coordinate more than often require high level quantum mechanical descriptions beyond single-determinant (including Kohn–Sham) approximations, where most EDAs break down. We examine a few illustrative cases. The LiF molecule is a closed-shell system very well described at the Hartree–Fock level. Even in a simple SR description, Table 1 shows that the chemical bonding protocol selects the ionic reference if an EDA is performed. If the neutral fragments are considered to engage in electron sharing, a small stabilizing electrostatic energy appears, together with a non-negligible Pauli repulsion caused by the overlap of the Li $2s$ function with the bonding $2p_z$ orbital of F and a very large orbital stabilization. This picture is turned upside down if the ionic reference is used, which requires an experimental ionization cost of $45.9$ kcal mol$^{-1}$,

**Table 1 Energetic reference bias in LiF.**

| Reference | $\Delta E_{\text{elstat}}$ | $\Delta E_{\text{Pauli}}$ | $\Delta E_{\text{orb}}$ |
|---|---|---|---|
| LiF | −14.26 | 60.90 | −135.85 |
| Li$^+$F$^-$ | −183.83 | 33.96 | −26.62 |

HF//def2-SVP EDA analysis with neutral and ionic references. Energies in kcal mol$^{-1}$.

obtained as the sum of the first ionization energy of Li and the electron affinity of Fg[49]. Reference-less IQA data lead to electrostatic $V_{\text{cl}} = -173.9$ and covalent $V_{xc} = -32.9$ kcal mol$^{-1}$ contributions, respectively, in rather good agreement with their EDA equivalents for an ionic reference. This accord is easy to understand if we examine the distribution of electron counts in the LiF atoms: $p(2, 10) = 0.887$, $p(3, 9) = 0.095$, $Q(\text{Li}) = +0.910$. The distribution is overwhelmingly dominated by one electron count, and a single (ionic) reference suffices. The EDA protocol works well in this case, but only if we allow for non-neutral fragment references, i.e., if we admit the electron count bias. Dissociation along the $^1\Sigma^+$ ground state curve leads to neutral fragments through a very well-studied multireference avoided crossing, that needs at least a two-determinant wavefunction for correct modeling. In the LiF case, since the equilibrium and long distance regimes are dominated by clear electron counts, judicious chemical intuition allows to manually select those states that provide the best chemical bonding model. In more complex cases this is not easily done. Even in LiF, EDA cannot be used at intermediate distances to follow the change of reference, but a multiconfigurational character poses no problem for real space analyses. A MR calculation[51] shows that a tight avoided crossing between the S0 $1^1\Sigma^+$ and S1 $2^1\Sigma^+$ states, which become <2 kcal mol$^{-1}$ apart, occurs at $R \approx 6.6$ Å. As we decrease $R$ and pass the avoided crossing, all $E_{\text{self}}$ and $E_{\text{int}}$ components jump suddenly from the values expected for an electron-sharing interaction between neutral entities to those of an ionic interaction between Li$^+$ and F$^-$ ions, the contrary being true on the $2^1\Sigma^+$ curve (see Fig. 4 and Supplementary Note 5). This violent transition shows how the chemical bonding between two species can change character abruptly. It is remarkable that from dissociation down to almost the ground state equilibrium geometry the electron distribution is dominated by just two components,[52] $p(n_{\text{Li}} = 3, n_{\text{F}} = 9) = p$ and $p(n_{\text{Li}} = 2, n_{\text{F}} = 10) \approx 1-p$, and that their evolution in the S0 state is almost symmetric to that in S1, with $p_{\text{S0}} \approx 1-p_{\text{S1}}$. At the crossing, $p$ is 0.5. A crystal clear chemical bonding model emerges: bonding in LiF is extremely well parametrized by a 2c, 1e bond where the probability of the electron being found in one or the other atom (á la Feynman) changes from one to almost zero. In this case, the use of a physical energetic partition applicable during the complete bonding process, both in ground and excited states, validates the insights obtained from the two-point (equilibrium, dissociation) EDA protocol. In passing, we comment on the reticence extended among many researchers[2,53] about the existence of true ionic bonding outside ionic crystals, where ions are lattice-stabilized. This reluctance is not found among molecular physicists, well acquainted with avoided crossings. A chemical model of how the ionization barrier is overcome without a total energy barrier is as follows: two neutral atoms interact, attracting themselves so that their overall energy decreases. At about the avoided crossing, the energy of the ionic arrangement competes with the neutral one, so that the jump occurs in quasi-degeneracy conditions. At equilibrium, almost only one electron distribution dominates (the ionic one), and a CCSD/aug-cc-pVTZ calculation yields Li,F moieties with self-energies just 9 and 25 kcal mol$^{-1}$ above those of the Li$^+$ and F$^-$ ions[48].

*Limitations*. Let us consider the formation of CH$_4$ from a C atom and a H$_4$ fragment, all maintaining tetrahedral symmetry. Standard chemical wisdom proposes to promote the C atom to the $^5$S–2s$^1$2p$^3$ state and then to hybridize in order to form four bonds. The two steps need about 96.5 and 62 kcal mol$^{-1}$, respectively[48]. A MR calculation[48] showed that as $R_{\text{C–H}_4}$ decreases, the probability of finding six electrons in C (and with it the usefulness of a neutral C reference) also decreases. At equilibrium, $p(n_{\text{C}} = 6)$ is about 0.3, and all $p(n_{\text{C}})$ with $n_{\text{C}}$ ranging from 2 to 10 are non-negligible. If only the $n_{\text{C}} = 6$ components are considered, the weight of a quintet contribution peaks at a maximum of about 20% at a $R_{\text{C–H}}$ distance of about 1.9 Å. This makes just a 6% of all the components. Assuming a fixed electron count for atoms is not justified. At variance with the avoided crossing case, where one distribution is very quickly changed into another, no abrupt change of any descriptor is found along the ground state energy curve, since the weights of all distributions change smoothly. Thus, as the fragments progressively interact and deform, $\Delta E_{\text{prep}}$ for the statistical mixture of electron distributions and electronic/spin states evolves also smoothly. At equilibrium, $E_{\text{self}}^{\text{C}}$ is 62 kcal mol$^{-1}$ above the C triplet ground state, which is much lower than the expected 158 kcal mol$^{-1}$ of the EDA promotion+hybridization energies. Transit from dissociation to equilibrium occurs, thanks to statistical mixing, through C atoms that never exhibit the full preparation energy of the EDA modeling. The usefulness and predictive ability of the EDA protocol is not challenged with this discussion: it is simple and easy to use. We just show that reference-less formalisms uncover a more complex reality, and that care should be taken in applying EDA blindlessly.

Another example follows the wake of the C$_2$F$_4$ case: the formation of the triple bond in acetylene from the interaction of two methylidyne CH radicals, which display $^2\Pi$ ground states 17.3 kcal mol$^{-1}$ below $^4\Sigma^-$ quartets. As expected, the EDA $\Delta E_{\text{orb}}$ is smaller when the partitioning is performed from two quartet-prepared CH fragments. Also, and this shows very nicely the predictivity of the EDA protocol, the minimum energy path for the association reaction is non-collinear, proceeding via a *trans* configuration of the doublet CH moieties until at about a C–C distance close to 1.35 Å an abrupt linearisation of the fragments occurs (see Fig. 5). A spatial decomposition shows an interesting story (Supplementary Note 6). At about $R_{\text{C–C}} = 2.6$ Å, when the CH fragments stop their free rotation adopting a clear trans geometry, $E_{\text{self}}^{\text{CH}}$ suffers a rather sudden increase, staying about 24 kcal mol$^{-1}$ above the isolated CH ground state energy for a wide range of inter-fragment distances down to linearisation, where it grows again. This behavior is clearly compatible with a quartet promotion. The figure also shows the contributions of the doublet and quartet states for the (7, 7) neutral fragment distribution, whose weight evolves from 1.0 at dissociation to about 0.4 at equilibrium. A comparison with dinitrogen, where the atomic fragments are already in prepared quartets at dissociation, is also displayed. For acetylene, a sudden increase in the weight of the quartet is found coupled with the behavior of $E_{\text{self}}^{\text{CH}}$. As the figure shows, linearisation coincides with coalescence of the electron distribution probabilities over those of dinitrogen. Thus, physically rigorous reference-less methods provide bonding models in agreement with those of successful, yet heuristic approaches. They also helps us understand when, how, and to what extent the use of fixed references will succeed or will be doomed to fail.

*References and dative bonding*. Dative bonds break heterolytically, being usually weaker than electron-sharing ones, that break homolytically. However, as the C$_2$F$_4$ and C$_2$H$_2$ examples show, if the nature of the chemical interactions changes along the dissociation channel, it is in the end a set of EDA calculations with

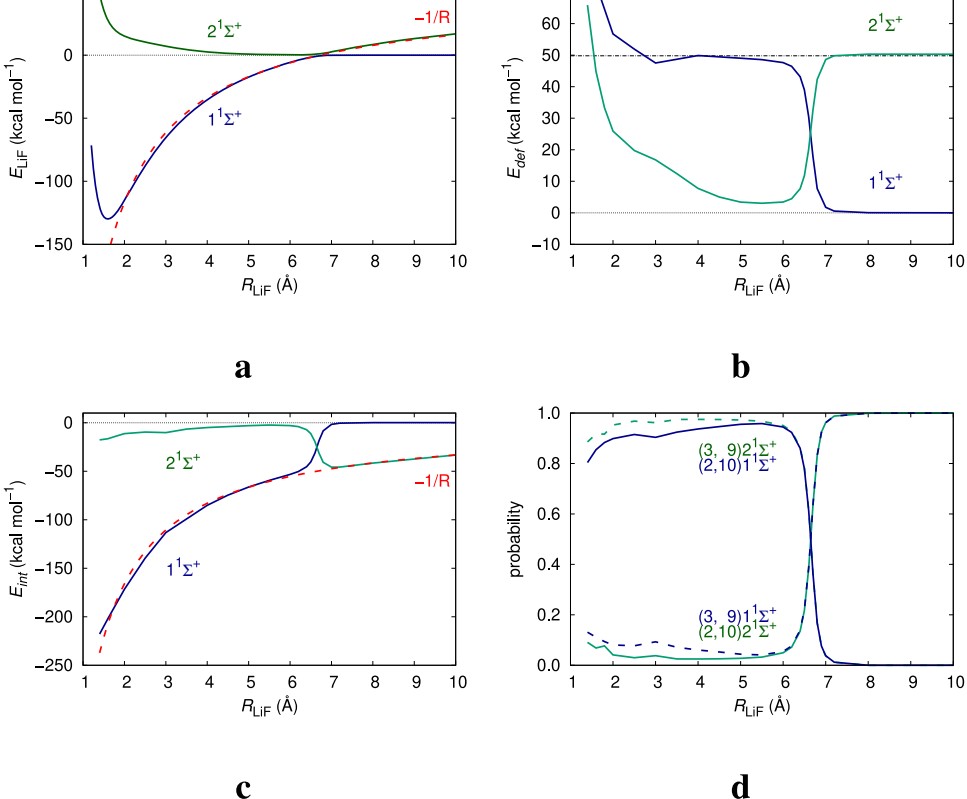

**Fig. 4 Spatial partitioning in LiF.** d-aug-cc-pVDZ MRCI-SD data in both the S0 ($1^1\Sigma^+$, dark-blue) and S1 ($2^1\Sigma^+$, dark-green) states along the dissociation curve. **a** Total energy, showing the avoided crossing (the $\approx 2$ kcal mol$^{-1}$ gap is barely visible at the scale of the plot), with the Coulombic $-1/R$ tail in red. **b** Total self-energy ($E_{self}^{Li} + E_{self}^F$). The energy reference is set to the neutral reference, while the ionization cost of LiF is 49.5 kcal mol$^{-1}$. **c** Total $E_{int}^{LiF}$, together with the Coulombic $-1/R$ contribution, in red. **d** Probabilities $p(n_{Li}, n_F)$ of the main electron distributions. The neutral (3,9) and ionic (2,10) partitions are plotted in dashed and full lines, respectively.

different fragment references which are used to decide between electron-sharing or donor–acceptor models. For instance, in the case of carbon suboxide, $C_2O_3$, the linear CO=C=CO model implies triplet CO fragments as well as a quintet carbon atom, a promotion that requires about 375 kcal mol$^{-1}$, while the angular CO → C ← CO one needs only exciting the C atom to its intra-configuration $^1$D state, with a much lower cost of 29 kcal mol$^{-1}$.[54] This points toward a promising role for reference-less methodologies in establishing the nature of dative bonding, that we illustrate with two systems. LiBe$^+$ dissociates into $^1$S Li$^+$ and Be. In a SR calculation the charge of the Li atom is 0.946 a.u., with $E_{int} = -30.2$ kcal mol$^{-1}$ dominated by a $-21.0$ kcal mol$^{-1}$ charge-dipole $V_{cl}$ term. The electron distribution has two main contributions, $p(2, 4) = 0.943$, $p(3, 3) = 0.054$, so a bonding model emerges in which only one of the two valence electrons of Be delocalizes (Supplementary Note 7). It is important to consider the two electrons of a seemingly normal 2c, 2e interaction independently. Using common spatial orbitals for them, as it is done in single or pseudo-single determinant approximations (e.g., in standard DFT) will mask an otherwise simple behavior, which is only recovered artificially (i.e the so-called left-right correlation) in multiconfigurational calculations or high-level valence bond approaches[55]. A simple solution is to use a (symmetry breaking) unrestricted single-determinant (UHF) ansatz. Figure 6 shows that the two 2s electrons of Be in LiBe$^+$ become spatially different at the UHF level: only one of the spatial orbitals delocalizes. A more interesting system is $^1\Sigma^+$ BeO, which dissociates to ground state Be and $^1$D excited O. MR data unveils how the interaction evolves from a donor-acceptor Be → O link to a very clear ionic-like

regime. Since O$^{2-}$ is not stable in vacuo, it is not possible to perform a standard EDA with charge-transfer Be$^{2+}$ and O$^{2-}$ references, a problem that doubtlessly biases interpretations. The BeO interaction (Supplementary Note 8) starts with an O atom displaying a kind of σ-hole, visible in the Laplacian of the density, which allows for a Be to O donor–acceptor σ favorable approach. The evolution of both the bond order, and particularly, the crystal clear behavior of $E_{self}^{Be}$ evidence that as $R$ decreases, a two-step ionization process occurs. In the first, similarly to what was found in LiBe$^+$, a Be electron is transferred to the oxygen. This is half completed at about $R = 2.6$ Å, and at around 2.4 Å the second ionization of Be starts. Simultaneously, already at 2.6 Å two back-bonding π bonds involving the oxygen's electrons develop (see Supplementary Note 8 and Supplementary Data 1). Thanks to back-donation, at equilibrium, $Q(Be) = 1.416e$, although the Laplacian, for instance, shows quasi-spherical atomic shells. Also, $p(2, 10) > 0.5$. After the second ionization starts, and surely at equilibrium, a better description of the system is that of an oxide anion donating rather symmetrically its σ and π electrons to a Be dication that acts as acceptor. In this regime, a dative, donor–acceptor bonding model is also admissible, but now the donor agent is the oxide anion.

Ubiquitous reference biases permeate the theory of chemical bonding, one of the pillars of modern chemistry. On the one hand, the internal references needed to interpret molecular wavefunctions in terms of atomic (i.e., chemical) components, lead to the interference terms that lie at the root of the standard model of the chemical bond. On the other, most energy decomposition analyses, which become essential to discern

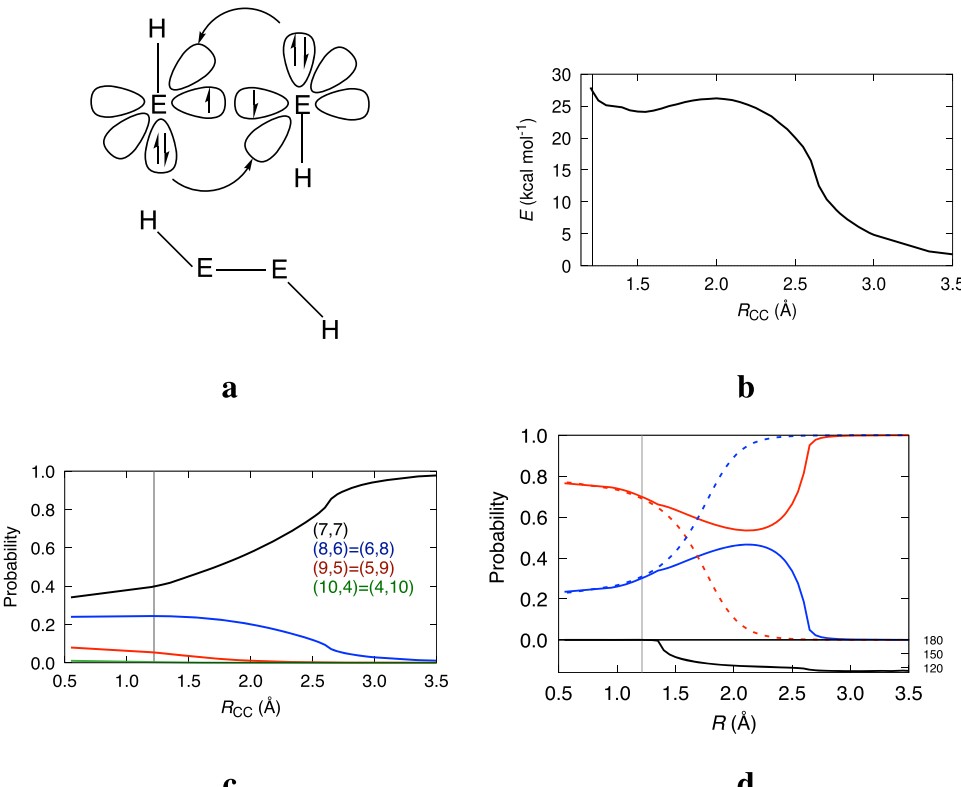

**a**                                    **b**

**c**                                    **d**

**Fig. 5 Spatial partitioning in acetylene.** CASSCF/def2-TVZPP data along the association process of two ground state $^2\Pi$ CH fragments to form acetylene. **a** Chemical bonding model explaining the trans geometry path. **b** $E_{\text{def}}$ of the CH fragment. **c** Probabilities of the $(7, 7)$, $(8, 6) = (6, 8)$, $(9, 5) = (5, 9)$, and $(10, 4) = (4, 10)$ electron partitions in black, blue, red, and green, respectively. **d** Normalized probabilities for the $^2\Pi$ and $^4\Sigma$ spin states of the $(7, 7)$ CH fragment neutral distribution (red and blue lines), respectively. The equivalent probabilities for the $^2D + ^2S$ and $^4S$ states obtained for $N_2$ at the same level of theory are also drawn with the same colors but with dashed lines. The variation of the HCC angle along the process is found in the bottom panel. Vertical lines are drawn at the equilibrium distance.

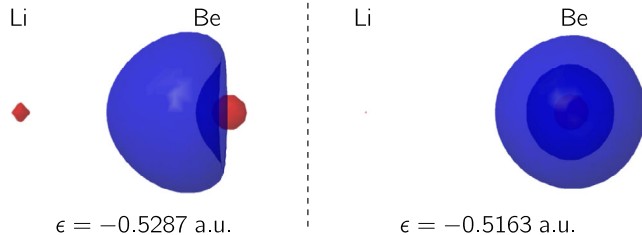

Li            Be              Li            Be

$\epsilon = -0.5287$ a.u.          $\epsilon = -0.5163$ a.u.

**Fig. 6 Symmetry broken LiBe$^+$.** Be $2s$-like orbitals in LiBe$^+$ at the UHF/6-311G(p) level, computed at $R_{\text{LiBe}} = 2.6$ Å. Orbital energies are also depicted. Only the left function is actively delocalized over the Li moiety. The isosurfaces shown correspond to $|\phi| = 0.1$ a.u.

among conflicting or alternative bonding images, rest on choosing external (energetic) references for the set of fragments that are meant to interact with each other. However, interacting fragments in-a-molecule are entities in pesudo-mixed quantum states, with fluctuating (i.e., not fixed) electron populations, electronic, and spin states. Failing to consider this fact leads to a variety of electron count or electron spin biases. The historical use of energy-promoted and/or spin-excited fragments to rationalize the nature of chemical links finds a rigorous conceptual root in considering fragments as open quantum systems. We have argued that real space analyses are as reference-less as possible, needing only the specification of a chemically sensible atomic partition of the space. If that ingredient is provided, no biases remain, and all the electron counts, electronic and spin states of the interacting

fragments are part of the output, not of the input of the procedure. In this way, the best bonding model for a system is automatically read from the results of the analysis.

## Methods
Electronic structure, QTAIM/IQA, and electron distribution calculations were performed with GAMESS[56], PROMOLDEN[57], and EDF[58], respectively. All atomic charges and probabilities cited were obtained with QTAIM atoms. Details of calculations: $H_2^+$, FCI/cc-pVTZ; $H_2$, CAS[2,2]//6-311G(p); LiNa$^+$, HF//6-311G(p,d) $R_{\text{eq}} = 3.460$ Å; $He_2^+$, FCI//cc-pVTZ, $R_{\text{eq}} = 1.332$ Å; LiF single reference, HF/def2-SVP, $R_{\text{eq}} = 1.54$ Å; LiF multirreference, MRCI-SD/d-aug-cc-pVDZ; $CH_4$, $C_2H_2$, CASSCF/def2-TZVPP; LiBe$^+$, HF//6-311G(p), $R_{\text{eq}} = 2.646$ Å, UHF also at the same geometry; BeO, CASSCF[8,8]/6-311G(p), $R_{\text{eq}} = 1.345$ Å;

## Data availability
The data generated in this study are provided in the Supplementary Information. Supplementary Note 1 contains a summary of the standard model of the chemical bond in $H_2^+$ and $H_2$. Supplementary Note 2, a primer on real space chemical bonding. Supplementary Notes 3–8 contain the data supporting the claims made for $H_2^+$, the $He_2^{2+}$ ion, the LiF molecule, the dissociation process of acetylene, LiBe$^+$ and BeO molecules, respectively. Supplementary Data 1 reports raw IQA and EDF data in BeO.

## Code availability
The GAMESS and EDF codes are available from refs. [56],[58]. The PROMOLDEN code is available from the authors upon request at ampendas@uniovi.es.

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

## Acknowledgements

Financial support from the spanish MICINN, grant PGC2018-095953-B-I00, and the FICyT, grant IDI/2021/000054, both awarded to A.M.P., is acknowledged.

## Author contributions

E.F.M. and A.M.P. contributed equally to this work.

## Competing interests

The authors declare no competing interests.
