## [Peer Review File · Nature Communications]

Comments on the paper “The role of references and the elusive nature of the chemical bond” by Pendás and Francisco

I thoroughly enjoyed reading this spirited paper, which discusses some basic features of the nature of the chemical bond that make an important contribution to the present understanding of chemical bonding. I want to point out that my personal point of view is somewhat different to those of the authors, who advocate real-space partitioning methods, where they made fundamental and ground-breaking contributions. My own viewpoint is that from a wave function-based method which is complementary and in my point of view more fundamental but it also has some disadvantages. I think this paper should be read by anyone currently involved in understanding chemical bonding, as it provides a clear perspective of the future. I have some comments given below which might be considered in a revised version.

1. The authors correctly point out at the bottom of page 7 that the electron-pair bond is not a result of electron spin pairing. They are saying on page 8 that Head-Gordon et al have questioned this view and rather suggested that electron delocalization as the most general bonding driver. They might add that Head-Gordon concluded this by considering the electronic triplet state of H_2 as reference, which is clearly not an appropriate reference for the chemical bonding in H_2 .
2. The authors discuss at the end of their section the one-electron bond of the alkali dimer cations giving Li_2^+ and $LiNa^+$ as examples. In the following section they discuss the H_2 molecule as a two-electron system. It would be interesting to include Li_2 in the section of the electron-pair bonding, because it exhibits the peculiar finding that the two-electron bond is weaker than the one-electron bond in Li_2^+ .
3. The authors are saying on top of page 10 that the EDA has problems with molecules that display exothermic dissociations. I do not quite understand this. The EDA gives in such cases the interaction energy which makes the molecule to be unstable, because it is considered only in the equilibrium geometry which is higher energy than the fragments. The energy terms of the EDA can only be related to thermodynamic stability but not kinetic stability. Why is this problem?
4. I also do not quite understand the criticism on the EDA treatment of LiF using charged or neutral species. Similarly as in C_2F_4 , the two curves of the energy terms can be drawn, which would show that the orbital interaction indicates regions where the neutral or the charged fragments are better descriptors. I also want to point out that figure 4a does not really show the avoided crossing, because the present version does show a crossing.
5. There are some typos and minor points:
 - a. Figure 2 should be redrawn. The long-dashed lines are not really recognized.
 - b. Typos on page 12, line 286 “fist”, page 13, line 293 “multirreference”, page 13, line 291, the sentence starting with “The EDA protocol works...” is not readable.
 - c. Page 20, reference 13: it should be a large C and a subscripted 2.
 - d. Concerning the references on C_2 (13-15) the following work should also be cited: M. Hermann, G. Frenking, *Chem. Eur. J.* **2016**, *22*, 4100.

Reviewer #1 (Remarks to the Author):

These are suggestions and minor corrections and I do not need to see the revised version. I strongly recommend acceptance of this paper in *Nature Communications*.

I prefer to sign the report with my name: Gernot Frenking

Reviewer #2 (Remarks to the Author):

This paper addresses the questions of the nature of chemical bonds, as usually interpreted from various references and through the concept of wave function interferences. The authors recommend the use of "real space analyses", which are free of reference choices, or as much free as possible.

They illustrate their criticisms of usual internal references on the exemple of the dihydrogen cation. They beautifully demonstrate the arbitrary nature of wave function interface in this case, by noting the absurdity of considering any interference between two events that can never be simultaneously present.

Same considerations are extended to 2-electron bonds.

This thought-provoking paper is welcome as it encourages chemists to avoid concepts that are not based on rigorous concepts.

I recommend publication without changes.

Reviewer #3 (Remarks to the Author):

This Manuscript by Ángel Martín Pendás and Evelio Francisco is a nice review article on several different theoretical approaches and philosophies to describe the chemical bond. In general, I found the article well written and easy to read. I seriously have no issues with it, except one major concern: This is in fact a very good review article, but I believe there is really nothing new, nor revolutionary, worth of publication on Nature Communications. In addition, I believe the manuscript to be too long as a "communication", and I think it will be hardly possible to condense it to a format that is in line with the scope of this journal. The real-space analysis based on AIM fragments is really the only portion that has some innovative character, but to a further examination, it is not substantially different from other AIM analysis.

Again, I don't see any flaws in the analysis, interpretation, and conclusions, the methodology is sound, the work is reproducible, and no additional evidence would be needed. However, there is really no noteworthy results, and it does not represent a significant contribution to the field, other than as a well-written review piece. As such, I cannot recommend it for publication onto "Nature Communications" but I think it should find a more suitable journal as a review article in computational chemistry.

Answer to Reviewer #1

I thoroughly enjoyed reading this spirited paper, which discusses some basic features of the nature of the chemical bond that make an important contribution to the present understanding of chemical bonding. I want to point out that my personal point of view is somewhat different to those of the authors, who advocate real-space partitioning methods, where they made fundamental and ground-breaking contributions. My own viewpoint is that from a wave function-based method which is complementary and in my point of view more fundamental but it also has some disadvantages. I think this paper should be read by anyone currently involved in understanding chemical bonding, as it provides a clear perspective of the future. I have some comments given below which might be considered in a revised version.

- 1. The authors correctly point out at the bottom of page 7 that the electron-pair bond is not a result of electron spin pairing. They are saying on page 8 that Head-Gordon et al have questioned this view and rather suggested that electron delocalization as the most general bonding driver. They might add that Head-Gordon concluded this by considering the electronic triplet state of H_2 as reference, which is clearly not an appropriate reference for the chemical bonding in H_2 .*

We agree with this, and have introduced a warning in the line suggested by the reviewer on page 8.

- 2. The authors discuss at the end of their section the one-electron bond of the alkali dimer cations giving Li_2^+ and $LiNa^+$ as examples. In the following section they discuss the H_2 molecule as a two-electron system. It would be interesting to include Li_2 in the section of the electron-pair bonding, because it exhibits the peculiar finding that the two-electron bond is weaker than the one-electron bond in Li_2^+ .*

The reviewer poses again a very interesting system in which traditional ideas break. However, since some of the editorial suggestions regarding the abstract and the introduction have already implied to increase the length of the manuscript, we prefer not to add another probably lengthy discussion on Li_2^+ . We will surely include one in future publications.

- 3. The authors are saying on top of page 10 that the EDA has problems with molecules that display exothermic dissociations. I do not quite understand this. The EDA gives in such cases the interaction energy which makes the molecule to be unstable, because it is considered only in the equilibrium geometry which is higher energy than the fragments. The energy terms of the EDA can only be related to thermodynamic stability but not kinetic stability.*

Why is this problem?

The sentence to which the reviewer refers to was not fortunate. It has been deleted. Provided that the He_2^{2+} metastable case is analysed in depth in the paragraph following that sentence, where the results of both EDA and real space decompositions are compared, we think that after this deletion it is clear that in these cases ΔE_{orb} can be related to covalency much as V_{xc} . We thank the reviewer for noticing this.

4. *I also do not quite understand the criticism on the EDA treatment of LiF using charged or neutral species. Similarly as in C_2F_4 , the two curves of the energy terms can be drawn, which would show that the orbital interaction indicates regions where the neutral or the charged fragments are better descriptors. I also want to point out that figure 4a does not really show the avoided crossing, because the present version does show a crossing.*

We partially understand the reviewer. After re-reading the paragraphs on LiF, we believe that our points are relatively clear, but we have added a paragraph on page 13 to clarify them more. The point is that in LiF it is easy to recognise the two possible electron counts that can be used to perform an EDA (the neutral and ionic ones). In general cases, where spin states mix with different possible ionisation channels, it is not obvious what fragmentation to use, so that one depends on chemical intuition. Even more, if one tries to follow the bonding formation (or dissociation process) to help us select the appropriate fragmentation channel, as soon as the system becomes multiconfigurational (as in LiF), EDA cannot help.

As the second point is regarded, the two states in Fig 4a do really not cross although they seem to do so. At the avoided crossing, the states are separated by just 2 kcal/mol, so that the gap is barely visible. We have indicated that in the caption.

5. *There are some typos and minor points:*

- Figure 2 should be redrawn. The long-dashed lines are not really recognized.*
- Typos on page 12, line 286 "fist", page 13, line 293 "multirreference", page 13, line 291, the sentence starting with "The EDA protocol works..." is not readable.*
- Page 20, reference 13: it should be a large C and a subscripted 2.*
- Concerning the references on C2 (13-15) the following work should also be cited: M. Hermann, G. Frenking, Chem. Eur. J. 2016, 22, 4100.*

These are suggestions and minor corrections and I do not need to see the revised version. I strongly recommend acceptance of this paper in Nature Communications.

a) Figure 2 has been redrawn, increasing the width of the long-dashed line so that it is perfectly visible. b) Typos have been corrected. c) Corrected. d) The work by M. Hermann, G. Frenking (Chem. Eur. J. 2016, **22**, 4100) is cited in the revised version.

I prefer to sign the report with my name: Gernot Frenking

We thank Prof. Frenking for wishing not to remain anonymous.

Answer to Reviewer #2

This paper addresses the questions of the nature of chemical bonds, as usually interpreted from various references and through the concept of wave function interferences. The authors recommend the use of "real space analyses", which are free of reference choices, or as much free as possible.

They illustrate their criticisms of usual internal references on the example of the dihydrogen cation. They beautifully demonstrate the arbitrary nature of wave function interface in this case, by noting the absurdity of considering any interference between two events that can never be simultaneously present.

Same considerations are extended to 2-electron bonds.

This thought-provoking paper is welcome as it encourages chemists to avoid concepts that are not based on rigorous concepts.

I recommend publication without changes.

We are very glad that the reviewer finds our article welcome and appreciate very much the praise that he/she makes of it when expressing the opinion that it is thought-provoking and encourages chemists to avoid concepts that are not based on rigorous concepts.

Answer to Reviewer #3

This Manuscript by Ángel Martín Pendás and Evelio Francisco is a nice review article on several different theoretical approaches and philosophies to describe the chemical bond. In general, I found the article well written and easy to read. I seriously have no issues with it, except one major concern: This is in fact a very good review article, but I believe there is really nothing new, nor revolutionary, worth of publication on Nature Communications. In addition, I believe the manuscript to be too long as a "communication", and I think it will be hardly possible to condense it to a format that is in line with the scope of this journal. The real-space analysis based on AIM fragments is really the only portion that has some innovative character, but to a further examination, it is not substantially different from other AIM analysis.

Again, I don't see any flaws in the analysis, interpretation, and conclusions, the methodology is sound, the work is reproducible, and no additional evidence would be needed. However, there is really no noteworthy results, and it does not represent a significant contribution to the field, other than as a well-written review piece. As such, I cannot recommend it for publication onto "Nature Communications" but I think it should find a more suitable journal as a review article in computational chemistry.

We do really appreciate that the reviewer considers our article nice, well written and easy to read. However, we disagree with their opinion that '*there is really nothing new, nor revolutionary, worth of publication on Nature Communications*'. Whether a piece of research offers revolutionary results or it simply provides beautiful, interesting and/or useful results to the scientific community is, to some extent, a matter of taste and debate. The authors neither know for sure if their work is revolutionary or not, but what are rather sure to have examined in the manuscript some relevant points that are a source of continuous debate in the chemical bonding community.

To place our work in context we have found it necessary to describe in minimal detail what the state of the art in the topic is, but after this minimal introduction, our points are all but a review of known results. Much on the contrary, we convincingly show that several points that are taken for granted by most researchers are not at all obvious, and that a non-biased point of view, as that provided by our real space description, leads to question many of these tenets. We try in a way to be provocative, to elicit opinions (even if they are contrary to ours), and to encourage the chemical bonding community to think again about the deep root or nature of chemical bonding, which is at the very heart of chemistry, on which so much ink has been spilled, and about which our understanding is still limited.

We think that the point of view that we advocate (the real space perspective) contributes significantly to a better understanding of the nature of the chemical bond, and feel that a wide scope journal such as *Nat. Commun.*, is an appropriate place to publish our results.